# Clinical characteristics of enteric fever and performance of TUBEX TF IgM test in Indonesian hospitals

Syndi Nurmawati[1,2☉], Anggraini Alam[1,3☉], Hofiya Djauhari[1,2], Tuti P. Merati[1,4], Pratiwi Sudarmono[1,5], Vivi Setiawaty[1,6], Dona Arlinda[1,7], Retna Indah Sugiyono[1,7], Mansyur Arief[1,8], Usman Hadi[1,9], Abu Tholib Aman[1,10], Dewi Lokida[1,11], M. Hussein Gasem[1,12], Emiliana Tjitra[1,7], C. Jason Liang[1,13], Aaron Neal[1,13‡], Herman Kosasih[1‡], Muhammad Karyana[1,7‡], Chuen-Yen Lau[1,13‡], Bachti Alisjahbana[1,2,14‡]*

**1** Indonesia Research Partnership on Infectious Diseases (INA-RESPOND), Central Jakarta, Indonesia, **2** Research Center for Care and Control of Infectious Disease, Universitas Padjadjaran, Bandung, Indonesia, **3** Pediatric Department, Faculty of Medicine, Universitas Padjadjaran/Dr. Hasan Sadikin Hospital, Bandung, Indonesia, **4** Faculty of Medicine, Udayana University, Denpasar, Indonesia, **5** Faculty of Medicine, Universitas Indonesia, Central Jakarta, Indonesia, **6** Prof. Dr. Sulianti Saroso Infectious Disease Hospital, North Jakarta, Indonesia, **7** Health Policy Agency, Ministry of Heath, Central Jakarta, Indonesia, **8** Dr. Wahidin Sudirohusodo Hospital, Makassar, Indonesia, **9** Department of Internal Medicine, Faculty of Medicine, Universitas Airlangga/Dr. Soetomo General Academic Hospital, Surabaya, Indonesia, **10** Dr. Sardjito General Hospital/Faculty of Medicine, Universitas Gadjah Mada, Sleman, Indonesia, **11** Tangerang General Hospital, Tangerang, Indonesia, **12** Dr. Kariadi General Hospital/Faculty of Medicine, Universitas Diponegoro, Semarang, Indonesia, **13** National Institute of Allergy and Infectious Disease, National Institutes of Health, Bethesda, Maryland, United States of America, **14** Department of Internal Medicine, Faculty of Medicine, Universitas Padjadjaran/Dr. Hasan Sadikin Hospital, Bandung, Indonesia

☉ These authors contributed equally to this work.
‡ AN, HK, MK, CYL and BA also contributed equally to this work.
* b.alisjahbana@unpad.ac.id, b.alisjahbana@gmail.com

**Data Availability Statement:** All relevant data are within the manuscript and its Supporting Information files.

## Abstract

### Background

Accurate diagnosis of enteric fever is challenging, particularly in low- and middle-income countries, due to the overlap of clinical and laboratory features with other pathogens. To better understand the difficulties in enteric fever diagnosis, we evaluated the characteristics of patients clinically diagnosed with enteric fever and the real-world performance of TUBEX TF, one of the most used tests in Indonesia.

### Methodology/Principal findings

Patients were recruited through the AFIRE (Etiology of Acute Febrile Illness Requiring Hospitalization) study at eight Indonesian hospitals. Blood culture was performed for all patients, and TUBEX TF was performed for suspected enteric cases. *Salmonella* PCR and ELISA tests were performed at a reference lab. Sensitivity and specificity of TUBEX TF and IgM and IgG anti-*S.* Typhi ELISA were determined. Of 301 patients clinically diagnosed with enteric fever, 50 (16.6%) were confirmed by blood culture and/or PCR. Confirmed cases were mostly school-aged children presenting with fever, anorexia, dizziness and/or

**Funding:** This project has been funded in whole or in part with MOH Indonesia and Federal funds from the National Institute of Allergy and Infectious Diseases, National Institutes of Health, under contract Nos. HHSN261200800001E and HHSN261201500003I. The content of this publication does not necessarily reflect the views or policies of the Department of Health and Human Services, nor does mention of trade names, commercial products, or organizations imply endorsement by the U.S. Government.

**Competing interests:** The authors have declared that no competing interests exist.

abdominal pain with normal leukocyte count or leukopenia. TUBEX TF demonstrated a sensitivity of 97.6% to 70.7% and specificity of 38.3% to 67.2% at cutoffs of 4 and 6, respectively. Acute IgG demonstrated the best sensitivity and specificity, at 90.7% and 82.7%, respectively, and the best ROC characteristics.

## Conclusions/Significance

A substantial proportion of enteric fever was misdiagnosed at all study hospitals, likely due to the overlap of clinical characteristics and lab parameters with those of other common pathogens. The TUBEX TF rapid serological assay demonstrated suboptimal performance in our setting and tended to over-diagnose enteric fever. The role of IgG from acute specimens for identification of enteric fever cases merits additional consideration.

### Author summary

Enteric fever is a major health problem in Indonesia. Blood culture, the gold standard for enteric fever diagnosis, is expensive and requires several days to produce a result, leading to diagnoses being made through rapid tests such as TUBEX TF in Indonesia and other LMICs. In a real-world setting, we found that TUBEX TF over-diagnosed enteric fever by more than 50% in comparison to blood culture and PCR. Over-diagnosis may result in over-treatment and the subsequent increase of antimicrobial resistance. Given the suboptimal performance of TUBEX TF, the use of readily available clinical and laboratory data to facilitate identification of enteric fever cases has been considered. In our study, the majority of enteric fever patients were school-aged children. Although non-specific, gradually increasing fever, gastrointestinal symptoms, and cough were often present; hematologic parameters did not distinguish enteric fever from other etiologies. Interestingly, diagnostic accuracy was better by *S*. Typhi IgM and best by *S*. Typhi IgG in acute specimens. These findings may reflect the inaccuracy or subjectivity of TUBEX TF. The surprisingly excellent performance of *S*. Typhi IgG ELISA merits further evaluation.

## Introduction

Enteric fever is a global health problem, especially in low- and middle-income countries (LMIC). Studies suggest enteric fever causes 11–20 million infections and 120,000–200,000 deaths globally each year, with the preponderance of cases in South Central and Southeast Asia [1–4]. The annual rate of infection in Jakarta, the capital of Indonesia, is approximately 1.4 cases per thousand population [5]. In other parts of Indonesia, the annual incidence is estimated to be 3–8 cases per thousand [6,7]. However, the actual disease burden is unclear due to diagnostic challenges, and improving diagnostic accuracy would facilitate optimization of disease management [8].

Enteric fever is characterized by prolonged fever, disturbances of bowel function, headache, malaise, and anorexia [9]. Most cases are clinically indistinguishable from other locally prevalent etiologies of fever such as dengue fever and rickettsiosis infections [10]. Routine hematology tests such as leukocyte and neutrophil-lymphocyte count ratio (NLCR) may help differentiate viral and bacterial infections, but they are not specific. Blood culture is the gold standard for enteric fever diagnosis, but it is expensive [8], requires advanced laboratory

capability, and may take days to produce a result [11]. Moreover, sensitivity of blood culture can be low due to low levels of bacteremia [8,11] and widespread use of antibiotics, especially in LMIC [12]. Molecular detection by PCR may help in diagnosing enteric fever [11,13,14], but it has low sensitivity, is not widely available [15], and requires trained technicians [8].

Considering the challenges with blood culture and PCR, rapid serological tests such as TUBEX TF, Typhidot, and Test-it Typhoid tests are attractive options. However, they show only modest diagnostic accuracy [9,16]. Evaluation of these tools in Bangladesh and Tanzania by the WHO has demonstrated suboptimal clinical utility [17–19]. Thus enteric fever typically remains in the differential diagnosis for patients experiencing acute febrile illness, necessitating the use of empiric antimicrobial therapy [9,12].

TUBEX TF, a serological test detecting lipopolysaccharide (LPS) antibody IgM in *S*. Typhi, is the most common rapid test used to confirm clinically suspected enteric fever at hospitals in Indonesia. TUBEX TF is also considered a standard test for enteric fever by Indonesia's National Health Insurance, Jaminan Kesehatan Nasional. Due to this, the test was performed on patients suspected of having enteric fever. A meta-analysis showed an average sensitivity of 78% (95% CI 71% to 85%) and specificity of 87% (95%CI 82% to 91%) for TUBEX TF with a cut-off $\geq$ 3. The studies included were designed to assess TUBEX TF and therefore tended to overestimate the accuracy of test results [16]. A study from Sapkota et al. also revealed that TUBEX TF had a sensitivity and specificity of 60.6% and 94.0%, respectively [20]. TUBEX TF may not perform as well in the real-world Indonesian clinical setting given sample processing requirements and subjective aspects of colorimetric result interpretation.

This study aims to both describe the clinical characteristics of enteric fever patients enrolled in a prospective observational cohort study of patients hospitalized with febrile illness across seven Indonesian cities, and to evaluate the performance of TUBEX TF and IgM and IgG ELISA serology in a real-world Indonesian hospital setting using blood culture and PCR for *Salmonella* as reference tests.

## Methods

### Ethics statement

This study was approved by the institutional review boards of Dr. Cipto Mangunkusumo Hospital, Faculty of Medicine, Universitas Indonesia, and the National Institute of Health Research and Development (NIHRD), Ministry of Health, Indonesia. The study was performed in accordance with the Declaration of Helsinki. Study aims and procedures were explained to patients in Indonesian. Written informed consent was obtained from each adult subject or their legal guardian/representative. Informed consent in children aged 1–12-year-old was provided in writing by their parents or legal guardian. Children aged 13–17-years-old provided written assent. For illiterate subjects, the informed consent form was read to the subject in Indonesian by the investigators with a witness present. The consent form was subsequently signed, or a thumbprint accompanied by the witnesses' signature was provided.

Patients were recruited as part of the Etiology of Acute Febrile Illness Requiring Hospitalization (AFIRE) study, which was conducted in Indonesia from 2013 to 2016 at eight tertiary hospitals in seven cities. The study recruited patients who presented at the hospital sites with acute fever, were at least one year old, were hospitalized within the past 24 hours, and had not been hospitalized within the past three months. Clinical information and biological specimens were collected at enrollment, 14–28 days after enrollment, and three months after enrollment. Both standard of care (SOC) tests and study-specific tests were performed for participants, with SOC tests being performed at the direction of the attending physician in real-time to inform clinical care and study-specific tests being performed at a later time according to a

standard study testing algorithm. Initial clinical diagnoses, including enteric fever and non-enteric fever, were made based on clinical judgement alone and later modified based on diagnostic test results. Details of the AFIRE study and laboratory testing have been previously published [21].

## Aerobic blood culture

To identify etiologies of fever, acute blood samples from all subjects were cultured on-site in real-time according to SOC procedures. Blood was drawn using closed-system venipuncture within 48 hours of admission to the emergency department. Two BACTEC blood culture system bottles (Becton Dickinson Diagnostic Instrument Systems, Sparks, Md) or BacT/Alert bottles (BioMerieux, Inc., Durham, North Carolina) were prepared for each subject using 1–3 mL blood from each arm in pediatrics and 5–8 mL blood from each arm in adults. Aerobic bacterial cultures and automated growth identification were performed in accordance with the manufacturer's instructions. Blood culture results, including the distinction between *S*. Typhi and *S*. Paratyphi, were immediately provided to the attending physician to inform patient care decisions in accordance with SOC procedures.

## TUBEX TF test at local hospitals

Specimens from subjects with clinically suspected enteric fever were tested by TUBEX TF in real-time by hospital laboratory staff at the direction of the attending physician in accordance with SOC procedures. Specimens and accompanying clinical information, including provisional diagnosis, were provided to laboratory staff, and test results were immediately returned to the attending physician to inform patient care decisions. The TUBEX TF rapid typhoid detection assay (IDL Biotech AB, Karlsbodavägen 39, SE- 168 11 Bromma, Sweden) is a serological test kit that detects IgM antibodies to *S*. Typhi O:9 LPS antigen. Though *S*. Paratyphi is also detected due to high antibody cross-reactivity, the test is not able to distinguish the pathogens from each other. The TUBEX TF assay was performed following the manufacturer's instructions, where a score of 4 is considered weakly positive and 6–10 considered positive [22].

## Reference laboratory testing

Available study samples were sent to a central reference laboratory to be tested at a later date for *Salmonella spp*. and a variety of other pathogens known to cause acute febrile illnesses. Results were not obtained in real-time and were not used to influence patient care decisions. The methods and testing algorithm for those pathogens, particularly *Rickettsia spp.*, *Leptospira spp.*, dengue virus, and chikungunya virus, have been described previously [21]. Dengue tests (PCR, NS1, IgM, and IgG) were performed for all enrolled subjects at the reference laboratory, as described elsewhere [23]. Patients who were negative for dengue and other confirmatory tests at study sites were further tested for *S*. Typhi IgM and IgG antibodies using commercial ELISA kits (MyBiosource Cat. No. MBS580122 and MBS580121) according to the manufacturer's instructions. Both ELISAs measured antibodies against *S*. Typhi LPS, which is an antigen known to cross-react with antibodies against *S*. Paratyphi [24,25]. An antibody index < 0.9 was considered negative, 0.9–1.1 was considered borderline positive with follow-up testing to be performed as clinically indicated, and > 1.1 was considered positive. Seroconversion occurred when the index changed from negative in acute samples collected at enrollment to positive in convalescent samples collected during the 14–28 day visit window, or when IgM or IgG increased 2-fold between acute and convalescent samples.

  *S*. Typhi and *S*. Paratyphi were separately assessed by nested PCR from acute buffy coat specimens from patients diagnosed as non-Dengue and with a positive *Salmonella spp*.

serological assay from acute samples [13,26]. Briefly, pathogen genomic DNA was first obtained by processing buffy coat specimens with a QIAamp DNA Mini Kit (QIAGEN, Cat. #:51304 or Cat#: 51306) according to the manufacturer's instructions. A portion of the *fliC* gene of the *S*. Typhi was amplified using ST1 and ST2 primary PCR primers, followed by ST3 and ST4 nested PCR primers to produce a 343 base pair amplicon. A portion of the putative fimbrial gene in the *S*. Paratyphi was amplified using stkG_F1 and stkG_R1 primary PCR primers, followed by stkG_F2 and stkG_R2 nested PCR primers to produce a 229 base pair amplicon. Positive results were identified by visualization of appropriately sized PCR products through agarose gel electrophoresis. Positive results showed whether the blood sample contained *S*. Typhi or *S*. Paratyphi.

## Case definitions

Case definitions based on laboratory test results were developed for the purposes of data analysis. *Clinically diagnosed with enteric fever* cases were diagnosed with enteric fever on-site based on clinical presentation and/or initial laboratory examination. *Clinically diagnosed with non-enteric fever* cases were diagnosed with anything other than enteric fever on-site based on clinical presentation and/or initial laboratory examination. *Confirmed enteric fever* cases had at least one of the following: positive blood culture and/or PCR for *S*. Typhi or *S*. Paratyphi. *Probable enteric fever* cases had seroconversion/two-fold increase/high (Index > 1.1) *Salmonella* IgM and IgG antibodies, negative blood culture and PCR, and no evidence of infection by other pathogens (*Rickettsia spp.*, *Leptospira spp.*, dengue virus, and chikungunya virus) by PCR, immunofluorescence assay, microscopic agglutination test, NS1 antigen test, and/or serology according to a previously described algorithm [21]. *Undiagnosed* cases did not have evidence of *Salmonella* or other pathogens tested. In this study, enteric fever refers to infections by *S*. Typhi and *S*. Paratyphi A, as several studies have shown no significant difference between signs and symptoms of *S*. Typhi and *S*. Paratyphi A [10].

## Data analysis

Descriptive statistics were used to assess demographic, clinical and laboratory characteristics of subjects clinically diagnosed with enteric fever. Six age categories were used: pre-school (1–5 years old), school age (6–10 years old), adolescent (11–17 years old), early adulthood (18–25 years old), middle adulthood (26–40 years old), and late adulthood (41 or above). Timing of fever onset was based on patient self-report which was categorized into sudden or gradual according to patient's description. Gradual onset fever was defined as self-reported fever that rises slowly, while sudden onset of fever was defined as a sudden increase of temperature reported by patients. Median and interquartile ranges were calculated for continuous variables, and frequency and percentage were determined for categorical variables. Significance was assessed by the chi-squared test for each characteristic, with $p < 0.05$ being considered significant. In this study, we assessed characteristics of clinical presentation and serologic testing of subjects clinically diagnosed with enteric fever and performed a validity analysis for TUBEX TF and ELISA IgG and IgM.

## Clinical presentation of subjects clinically diagnosed enteric fever

The first analysis we performed was for subjects clinically diagnosed with enteric fever. In this analysis we compared characteristics, symptoms, and laboratory results between subjects with confirmed enteric fever, probable enteric fever, *Rickettsia spp.*, *Leptospira spp.*, dengue virus, and chikungunya virus. We also compared similar variables between subjects that were correctly and incorrectly diagnosed as enteric fever.

Among the subjects clinically diagnosed with enteric fever, we separated tests performed at the hospital and reference laboratory. We analyzed the mean days from symptom onset to testing for each assay while considering the proportion of positive results. We also evaluated the timing of positive results from blood culture, PCR, and TUBEX TF among patients with confirmed and probable enteric fever.

### Diagnostic accuracy

TUBEX TF sensitivity was determined from all confirmed enteric fever patients who had TUBEX TF testing as part of their hospital evaluation. Two TUBEX TF score categories, $\geq 4$ and $\geq 6$, were assessed since the manufacturer recommends interpreting a score of 4 as weakly positive and 6–10 as positive [22]. IgM and IgG ELISA sensitivities were determined from all confirmed enteric fever cases where specimens were available for testing at the reference laboratory. Each diagnostic test was additionally assessed with a receiver operating characteristic (ROC) curve, and the area under the curve (AUC) was calculated.

TUBEX TF specificity was determined from confirmed non-enteric fever cases where TUBEX TF testing was performed based on clinical suspicion. IgM and IgG ELISA specificities were determined from confirmed non-enteric fever cases where specimens were available for testing at the reference laboratory. Confirmed non-enteric fever cases in this analysis referred to cases that were found to be positive for *Rickettsia spp.*, Dengue, Chikungunya, or *Leptospira spp.* by blood culture, PCR, IFA, and/or MAT. Non-enteric fever cases that were positive by serology only were excluded.

Three sensitivity analyses of diagnostic accuracy were performed: first, stratification by site to check for heterogeneity of diagnostic accuracy; second, stratification by age ($<$18 years versus $\geq$18 years) to assess whether patient age affected diagnostic accuracy; and third, restriction to only patients with both TUBEX TF or ELISA results to perform a direct comparison of tests.

## Results

### Overall classification of cases

Of 1,486 subjects enrolled in the AFIRE study, samples from 22 participants were excluded due to mismatched specimens from one of our sites and 1,464 were eligible for this analysis (Fig 1). Sites clinically diagnosed 301 subjects with enteric fever and 1,163 with non-enteric fever. From the clinically diagnosed enteric fever group, 50 (16.6%) had confirmed enteric fever based on culture and PCR results, and 35 (11.6%) had probable enteric fever based on serological results. Most (75.5%) probable enteric fever cases were identified based on high acute IgM and/or IgG titers rather than seroconversion or twofold titer increases (S1 Table). Other etiologic pathogens were found in 138 (45.8%) of the remaining subjects clinically diagnosed with enteric fever, while no etiology was identified in 78 cases.

Of the 1,163 subjects clinically diagnosed with non-enteric fever, 366 did not undergo evaluation for enteric fever because they were already confirmed to have other pathogens. The remaining 797 subjects were tested for enteric fever, resulting in 4 cases of confirmed enteric fever by culture and 14 cases of probable enteric fever by serological tests (Fig 1). Among cases that were clinically diagnosed as non-enteric fever but later confirmed to be enteric fever, 2 were misdiagnosed as dengue fever, 1 as appendicitis, and 1 as acute limb ischemia. Among cases that were clinically diagnosed as non-enteric fever but later categorized as probable enteric fever, 3 patients were originally diagnosed with gastroenteritis, 3 with dengue fever, 2 with pneumonia, 1 with aseptic meningitis, 1 with viral infection, 1 with urinary tract infection and lung tuberculosis, 1 pharyngitis, 1 fever unknown, and 1 with Hirschprung's disease and cellulitis.

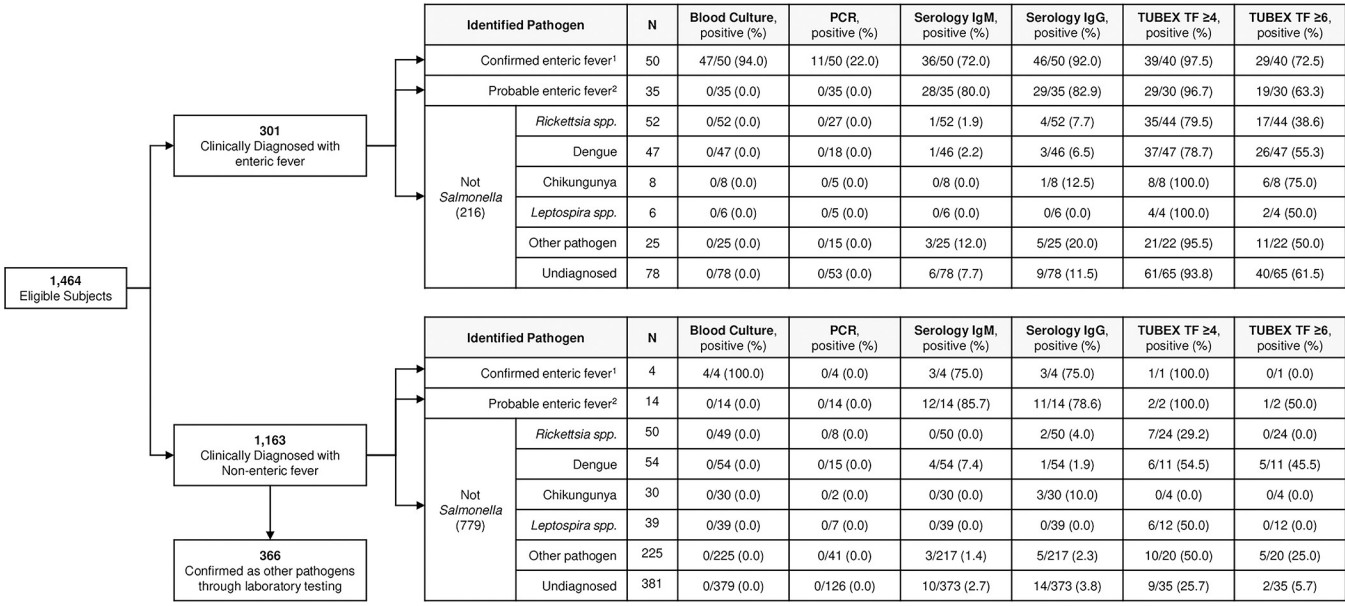

**Fig 1. Diagnostic flow and final etiological classification of cases from the AFIRE study.**

Finally, from both groups, a total of 103 enteric fever cases were identified, 54 (52.4%) of which were confirmed and 49 (47.6%) of which were probable. Among the 54 confirmed cases, 43 (79.6%) were confirmed by blood culture only, 3 (5.6%) by molecular detection only, and 8 (14.8%) by both blood culture and molecular detection. From those with positive PCR result, 1 was *Salmonella* Paratyphi and 10 were *Salmonella* Typhi. The 49 probable cases were classified based on serological assay results, most commonly high ELISA IgM and IgG antibody indices at baseline (24 cases, 49.0%). Details are shown in S1 Table.

## Clinical presentation of subjects clinically diagnosed as enteric fever

Subjects clinically diagnosed with enteric fever were evaluated as a group to understand potential overlap in demographics and clinical presentation. Table 1 shows a comparison of those subjects categorized as confirmed enteric fever, probable enteric fever, and other confirmed pathogens. A comparison between those correctly and incorrectly diagnosed with enteric fever is provided in S2 Table, while comparison between all enteric fever cases and other confirmed pathogen cases is provided in S3 Table.

Confirmed enteric fever cases were more often found in school age children (28.0%). In confirmed enteric fever cases, gradual onset of fever was observed in 29 subjects (58.0%), which differs significantly from subjects with confirmed *Rickettsia spp*. or dengue infections (32.7% and 31.9%, respectively). Significant differences were also observed between confirmed enteric fever and *Rickettsia spp*. cases for anorexia and diarrhea; confirmed enteric fever and dengue cases for abdominal pain, cough, and diarrhea; and confirmed enteric fever and chikungunya cases for abdominal pain, cough, and diarrhea. There were no significant differences between confirmed and probable enteric fever cases as seen in S1 Table. Complete blood count from confirmed enteric fever cases revealed that leukocyte count is often normal (74%), with lymphopenia (59.5%) and a median platelet count of $129.6 \times 10^3/mm^3$.

Comparison between correctly and incorrectly diagnosed enteric fever demonstrated that subjects correctly diagnosed as enteric fever had higher frequency of gradual onset of fever, anorexia, abdominal pain, headache, cough, and diarrhea (S3 Table).

**Table 1. Demographic and clinical characteristics of subjects clinically diagnosed with enteric fever categorized by the most common laboratory-confirmed final diagnoses from the AFIRE study.**

| | Confirmed Enteric Fever | Probable Enteric Fever | *Rickettsia spp.* | Dengue | Chikungunya | *Leptospira spp.* |
|---|---|---|---|---|---|---|
| **Total Positive Subjects, N** | 50 | 35 | 52 | 47 | 8 | 6 |
| **Demographics, N (%)** | | | | | | |
| Male subjects, N (%) | 29 (58.0) | 10 (28.6) | 35 (67.3) | 26 (55.3) | 4 (50.0) | 4 (66.7) |
| Age group, N (%) | | | | | | |
| 1–5 years | 8 (16.0) | 3 (8.6) | 3 (5.8) | 4 (8.5) | 0 (0) | 0 (0) |
| 6–10 years | 14 (28.0)* | 6 (17.1) | 1 (1.9)* | 6 (12.8) | 0 (0) | 0 (0) |
| 11–17 years | 8 (16.0) | 8 (22.9) | 8 (15.4) | 3 (6.4) | 0 (0) | 0 (0) |
| 18–25 years | 11 (22.0) | 12 (34.3) | 13 (25.0) | 20 (42.6) | 5 (62.5) | 1 (16.7) |
| 26–40 years | 7 (14.0) | 5 (14.3) | 8 (15.4) | 9 (19.1) | 0 (0) | 1 (16.7) |
| 41–98 years | 2 (4.0)* | 1 (2.9) | 19 (36.5)* | 5 (10.6) | 3 (37.5) | 4 (66.7) |
| **Signs/Symptoms at Enrollment** | | | | | | |
| Day of fever onset, median (IQR) | 7.0 (4.0–9.0) | 8.0 (6.8–11.0) | 7.0 (6.0–8.0) | 5.0 (3.0–6.0) | 2.0 (1.3–3.8) | 5.5 (3.0–6.5) |
| Gradual onset of fever, N (%) | 29 (58.0)*† | 24 (68.6) | 17 (32.7)* | 15 (31.9)† | 5 (62.5) | 3 (50.0) |
| Anorexia, N (%) | 26 (52.0)* | 20 (57.1) | 17 (32.7)* | 15 (31.9) | 2 (25.0) | 1 (16.7) |
| Abdominal pain, N (%) | 19 (38.0)†$ | 16 (45.7) | 12 (23.1) | 9 (19.1)† | 0 (0.0)$ | 0 (0.0) |
| Cough, N (%) | 20 (40.0)†$ | 17 (48.6) | 14 (26.9) | 9 (19.1)† | 0 (0.0)$ | 1 (16.7) |
| Diarrhea, N (%) | 20 (40.0)*†$ | 12 (34.3) | 7 (13.5)* | 9 (19.1)† | 0 (0.0)$ | 1 (16.7) |
| Constipation, N (%) | 10 (20.0) | 4 (11.4) | 16 (30.8) | 5 (10.6) | 1 (12.5) | 3 (50.0) |
| **Antibiotics Pre-hospitalization, N (%)** | 26 (52.0)*0 | 17 (48.6) | 12 (23.1)* | 16 (34.0) | 3 (37.5) | 0 (0.0)0 |
| **Hematology at Enrollment** | | | | | | |
| Hemoglobin (mg/dL), median (IQR) | 12.5 (11.3–13.8) | 12.5 (11.4–13.6) | 14.1 (13.0–15.2) | 13.8 (12.4–15.9) | 14.0 (13.1–14.8) | 13.1 (11.8–15.1) |
| Leukocyte (x1,000/mm³), median (IQR) | 6.0 (4.2–7.8) | 6.2 (4.8–8.3) | 6.9 (5.4–8.9) | 4.5 (3.5–6.2) | 5.8 (4.9–7.9) | 8.9 (6.4–10.2) |
| Lymphocyte (%), median (IQR) | 22.1 (15.7–35.0) | 30.9 (17.7–35.2) | 21.8 (18.8–26.0) | 22.6 (12.8–29.2) | 16.9 (12.6–22.9) | 7.2 (7.0–10.0) |
| Platelets (x1,000/mm³), median (IQR) | 129.6 (90.3–190.0) | 155.8 (112.0–223.0) | 109.5 (76.0–152.8) | 157.0 (103.0–185.0) | 195.0 (160.8–237.4) | 137.0 (90.3–244.0) |
| Granulocyte/Lymphocyte ratio, median (IQR) | 2.9 (1.6–4.6) | 2.0 (1.6–4.1) | 3.3 (2.5–4.1) | 2.8 (2.0–6.0) | 4.5 (2.5–6.1) | 11.2 (7.9–12.8) |

Notes: Significant (p < 0.05) differences observed

*between confirmed enteric fever and *Rickettsia spp*.

†between confirmed enteric fever and dengue, and

$between confirmed enteric fever and chikungunya.

0between confirmed enteric fever and *Leptospira spp*. There were no significant differences observed between confirmed and probable enteric fever.

A comparison of *S*. Typhi and *S*. Paratyphi A in S4 Table demonstrated that symptoms were similar, except for diarrhea being more prevalent in *S*. Typhi patients.

## Serologic testing

Sites conducted TUBEX TF in 260 (86.4%) of the clinically diagnosed with enteric fever group and 234 (90%) had a positive result. In the clinically diagnosed with enteric fever group, 216 cases were confirmed as non-enteric fever; however, TUBEX TF was positive in 166/190 tested (87.4%) using the cut-off ≥ 4 and 102/190 (53.7%) using cut-off point 6. TUBEX TF was also tested in 109 (9.4%) subjects of the clinically non-enteric fever group and 41 (37.6%) had a positive result. Details are shown in Fig 1.

**Table 2. Confirmatory test results from hospital sites and the reference laboratory for subjects clinically diagnosed with enteric fever categorized by the most common laboratory-confirmed final diagnoses from the AFIRE study.**

| | Confirmed Enteric Fever | Probable Enteric Fever | *Rickettsia spp.* | Dengue | Chikungunya | *Leptospira spp.* |
|---|---|---|---|---|---|---|
| Total Positive Subjects, N | 50 | 35 | 52 | 47 | 8 | 6 |
| **Confirmatory Tests at Hospital** | | | | | | |
| Day of testing, median (IQR) | 8.0 (7.0–9.0) | 10.0 (7.5–12.0) | 8.0 (7.3–9.8) | 6.0 (5.0–7.0) | 4.5 (2.3–7.0) | 7.0 (4.5–8.8) |
| TUBEX TF score, median (IQR) | 6.0 (5.0–6.0) | 6.0 (4.8–7.3) | 4.0 (4.0–6.0) | 6.0 (4.0–6.0) | 6.0 (5.3–7.8) | 5.0 (4.0–6.0) |
| TUBEX TF score ≥4, N (%) | 39/40 (97.5) | 29/30 (96.7) | 35/44 (79.5) | 37/40 (85.1) | 8/8 (100) | 4/4 (100.0) |
| TUBEX TF score ≥6, N (%) | 29/40 (72.5) | 19/30 (63.3) | 17/44 (38.6) | 26/40 (55.3) | 6/8 (75.0) | 2/4 (50.0) |
| Positive blood culture, N (%) | 47 (94.0) | 0 (0.0) | 0 (0.0) | 0 (0.0) | 0 (0.0) | 0 (0.0) |
| **Confirmatory Tests at Reference Lab** | | | | | | |
| Day of testing, median (IQR) | 8.0 (5.0–10.0) | 9.0 (7.0–11.0) | 8.0 (6.0–9.0) | 5.0 (4.0–7.0) | 3.0 (2.3–4.0) | 6.5 (4.0–7.5) |
| Positive acute ELISA IgM, N (%) | 30 (60.0) | 25 (71.4) | 1/44 (2.3) | 1 (2.1) | 0 (0.0) | 0 (0.0) |
| Positive acute ELISA IgG, N (%) | 45 (90.0) | 28 (80.0) | 5/44 (13.9) | 8 (17.0) | 0 (0.0) | 0 (0.0) |
| Positive PCR, N (%) | 11 (22.0) | 0 (0.0) | 0 (0.0) | 0 (0.0) | 0 (0.0) | 0 (0.0) |

Notes: Due to the small numbers, cases of other viral, other bacterial, and other pathogens were not analyzed in this table.

Median day of testing is defined as median day of illness before testing.

TUBEX TF and *Salmonella* IgM and IgG testing showed little difference in the average number of days from symptom onset to specimen collection for testing. Moreover, TUBEX TF test performed on non-enteric fever subjects revealed a high false positive rate in contrast with blood culture, PCR, ELISA IgM, and IgG results. These results are shown in Table 2.

Among the 103 confirmed and probable enteric fever cases, most subjects received tests during days 7–9 of illness. Positive blood cultures were more common in the first week of illness compared to the second week. TUBEX TF ≥ 4 showed consistently positive results across both weeks, though fewer tests were performed in the first week of illness. TUBEX TF ≥ 6 and *Salmonella* PCR performed better in the second week of illness. The highest positivity rates across all days were 64.0% for blood culture, 26.3% for PCR, 100% for TUBEX TF ≥ 4, and 83.3% for TUBEX TF ≥ 6. Results detailed in S1 and S2 Figs.

## Sensitivity and specificity analysis of TUBEX TF and ELISA IgG and IgM in confirmed enteric fever cases

TUBEX TF testing was performed for 41 out of 54 confirmed enteric fever cases and 134 cases with confirmed pathogens other than enteric fever. All confirmed enteric fever cases, as well as a subset of available cases with confirmed pathogens other than enteric fever, were tested for acute IgM and IgG by ELISA. Table 3 shows sensitivity and specificity results for serological assays performed for these subjects. The sensitivity of TUBEX TF was 97.6% and 70.7% when using scores ≥4 and ≥6, respectively. The specificity of TUBEX TF was 38.8% and 67.2% using scores ≥4 and ≥6, respectively. The sensitivity and specificity varied widely among sites, with hospitals in Bandung, Semarang, and Surabaya showing good performance (S5 Table). The sensitivity of IgM and IgG acute ELISA testing was 59.3% and 90.7%, respectively. The specificity of IgM and IgG acute ELISA testing was 95.5% and 82.7%, respectively. TUBEX TF performance in confirmed and probable enteric fever were similar as shown in S6 Table. Moreover, although TUBEX TF was regarded as the test for *S*. Typhi, our result showed not much difference in sensitivity between *S*. Typhi, *S*. Paratyphi, and *S*. spp (S7 Table).

IgG was the best predictor of confirmed enteric fever, followed by IgM and TUBEX TF (Fig 2, ROC curve). When stratified by age and clinical diagnosis as enteric fever vs. non-

**Table 3. Performance of TUBEX TF, IgM, and IgG ELISA tests on patients with a confirmed final diagnosis of either enteric fever or non-enteric fever.**

| Test | Sensitivity<br>Confirmed Enteric Fever Cases:<br>Number Positive / Number Tested<br>(%), [95% CI] | Specificity<br>Confirmed Non-Enteric Fever Cases:<br>Number Positive / Number Tested<br>(%), [95% CI] |
|---|---|---|
| TUBEX TF (cut-off $\geq$ 4) | 40/41 (97.6%)<br>[85.6–99.9] | 52/134 (38.3%)<br>[30.6–47.6] |
| TUBEX TF (cut-off $\geq$ 6) | 29/41 (70.7%)<br>[54.3–83.3] | 90/134 (67.2%)<br>[58.4–74.9] |
| Acute LPS IgG (cut-off >1.1) | 49/54 (90.7%)<br>[78.9–96.5] | 62/75 (82.7%)<br>[82.7–90.1] |
| Acute LPS IgM (cut-off >1.1) | 32/54 (59.3%)<br>[45.1–72.1] | 106/111 (95.5%)<br>[89.3–98.3] |

Notes: The sensitivity analysis examined the number of confirmed enteric fever cases that were also tested by TUBEX TF (n = 41) or ELISA IgG/IgM (n = 54). The specificity analysis examined the number of non-enteric fever cases confirmed by their corresponding standard test (culture or PCR or IFA or MAT) that were also tested by TUBEX TF or ELISA IgG/IgM. TUBEX TF was tested in 134 cases including *Rickettsia spp.* (n = 44), *Leptospira spp.* (n = 16), Dengue (n = 62), and Chikungunya (n = 12). ELISA IgG was tested in 75 acute samples including *Rickettsia spp.* (n = 31), *Leptospira spp.* (n = 8), Dengue (n = 26), and Chikungunya (n = 10). ELISA IgM was tested in 111 acute samples including *Rickettsia spp.* (n = 42), *Leptospira spp.* (n = 12), Dengue (n = 47), and Chikungunya (n = 10).

enteric fever, the kinetics of the IgG ELISA revealed that younger people have a greater rise in titers than older people when enteric fever is confirmed. Furthermore, titers at fever onset were comparable among younger and older enteric fever patients. Non-enteric fever controls maintained a constant, much lower titer level across timepoints. IgM ELISA results show similar kinetics, though of a lesser magnitude (Figs 3 and S3).

## Discussion

Enteric fever has major health and economic consequences in Indonesia, where approximately 20% of enteric fever patients require hospitalization [1,27]. The AFIRE study showed that among people hospitalized for acute febrile illness in Indonesia, approximately 10% of those with identified etiologies have enteric fever [21]. Although a definitive enteric fever diagnosis depends on blood culture or molecular confirmation [28], those methods are not commonly used due to high cost, time requirements, and low sensitivity [8,11,17,29]. Physicians are typically left diagnosing enteric fever based on clinical presentation and then treating empirically. If available, physicians commonly use rapid tests, such as TUBEX TF, to help diagnose enteric fever. TUBEX TF is included in the Indonesian national guidelines for enteric fever diagnosis and was performed on most patients with suspected enteric fever in the AFIRE study.

In our cohort of adults and children, we found that enteric fever cases are significantly more common among school-aged children. When characterizing clinical presentation across the cohort, we found that gradual onset of fever accompanied by gastrointestinal symptoms (anorexia, abdominal pain, and diarrhea) and cough were more frequent in enteric fever compared to other infections, which is consistent with other studies [10,30]. We did not identify any symptoms specific to enteric fever, nor did we identify pathognomonic hematology parameters as suggested by others [10].

We found that 216 cases (71.8%) of clinically diagnosed enteric fever were not caused by *Salmonella*, and 18 of 103 total cases (17.5%) of enteric fever were clinically misdiagnosed as non-enteric fever. Though some misdiagnoses are to be expected in resource restricted

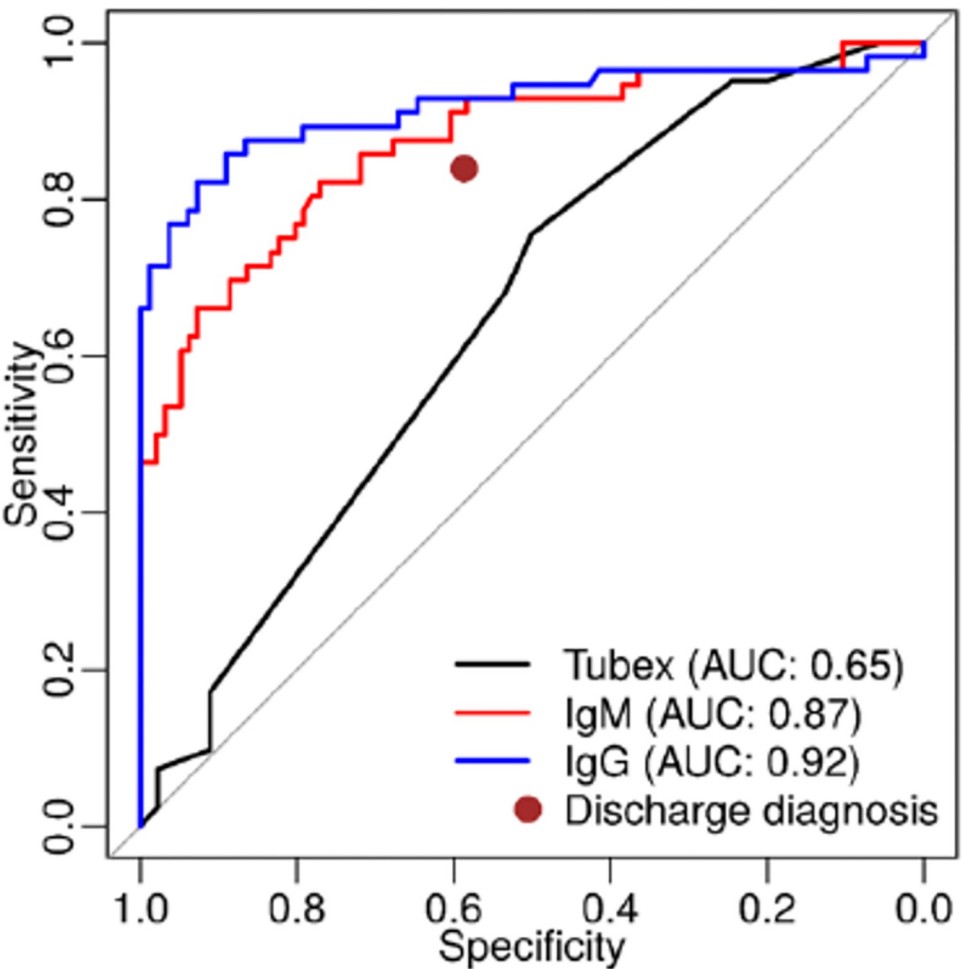

**Fig 2. ROC curve of clinical performance of clinical judgement compared to IgG ELISA, IgM ELISA, and TUBEX TF performed in acute samples in predicting laboratory-confirmed enteric fever cases.** Notes: Black line represents TUBEX test results performed on subjects on hospital sites during hospitalization. During 24 hours after hospital admission, blood was taken and tested for IgM (red line) and IgG (blue line) ELISA Salmonella. Red dot represents discharge diagnosis, defined as final diagnosis listed on the medical record on patient's discharge day.

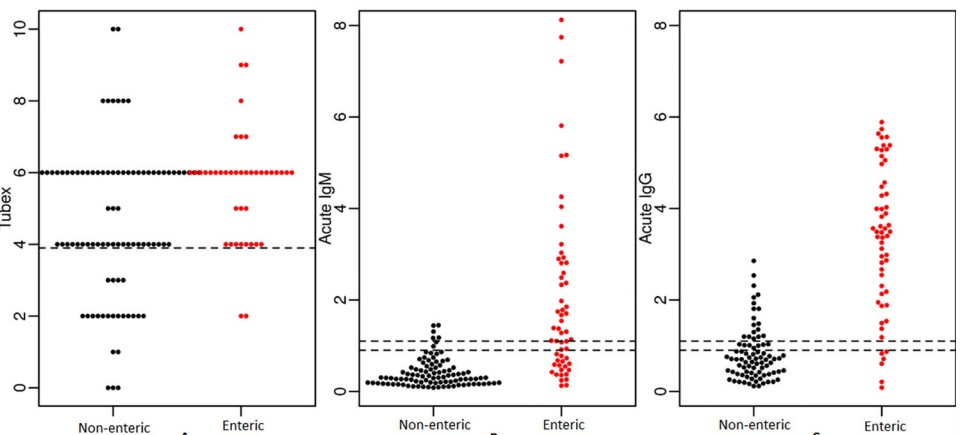

**Fig 3. Beeswarm plots of TUBEX TF (A), acute LPS IgM (B), and acute LPS IgG (C) in confirmed enteric fever and non-enteric fever cases.** Notes: Each spot represents one patient.

environments where testing is not readily available, our observations raise concerns for public health. Enteric fever misdiagnosis has been reported in other countries, though the numbers are lower than what we found in Indonesia [31]. Misdiagnosis is often attributed to poor performance and interpretation of rapid diagnostic tests such as Widal, Typhidot, and TUBEX TF, which are widely used in Indonesia [32,33]. However, the clinical misdiagnoses we observed were likely due to overlapping clinical presentations, highlighting a fundamental challenge of managing cases of acute febrile illness without sufficient access to high quality diagnostic testing. The underdiagnosis of enteric fever engenders inadequate antibiotic therapy and increased transmission, complications, and mortality [34,35], while overdiagnosis can lead to inappropriate antibiotic administration and increased antibiotic resistance [35].

The highest sensitivity (97.6%) was seen with TUBEX TF using a cut-off ≥4, but this was accompanied by the lowest specificity (38.8%). A cut-off score of ≥6 resulted in much higher, though still suboptimal, specificity (67.2%) at the expense of lower sensitivity (70.7%). Interestingly, our specificity results are lower than those previously reported [17,36,37]. Over half of clinically diagnosed enteric fever cases with positive TUBEX TF results were confirmed attributable to non-*Salmonella* pathogens. The subjective nature of operator interpretation, suggested by the highly variable specificity across sites, may have contributed to an overestimation of enteric fever cases. In contrast, only 2 cases of clinically diagnosed enteric fever that were non-*Salmonella* were found to be positive by IgM ELISA. The better performance of *S*. Typhi IgM ELISA compared to TUBEX TF may reflect the subjectivity or cross-reactivity of TUBEX TF. If the observed performance were attributable to population characteristics, we would expect IgM ELISA to perform similarly to TUBEX TF. Moreover, sensitivity and specificity of TUBEX TF varied between sites, unlike ELISA performance. This result emphasizes the subjectivity of grading positive results and the difficulties of this test compared to lateral flow assays. Performance of TUBEX TF might be improved with better standardization of reagent storage and assay techniques [38]. Incorporating TUBEX TF into a clinical algorithm to diagnose enteric fever may also improve its utility so long as it is not the sole diagnostic test [39].

The reference laboratory in our study made it possible for us to increase the rates of enteric fever diagnosis via PCR and ELISA. Our finding that the IgG ELISA test using acute specimens performed better than the IgM ELISA and TUBEX TF tests on the same specimens was surprising, particularly because IgG antibodies are considered to be an indicator of prior infection rather than acute illness. A study from Vietnam found that IgG against *S*. Typhi LPS was elevated in acute samples from culture-proven enteric fever patients compared to samples from hospitalized and community controls [40]. IgG levels from prior infection may wane and then exhibit a rapid anamnestic response with recurrent infection, obviating IgM production. This phenomenon was identified in several subjects where IgM antibodies were negative but IgG antibodies were elevated [41]. Since there are no reports about the kinetics of persistence of *S*. Typhi IgG antibodies, the significance of our ELISA IgG antibody results merits further investigation.

While our findings suggest that IgM and IgG ELISAs could be used as alternative methods for diagnosing enteric fever, costs, laboratory infrastructure requirements, and the need to do batch testing make them less feasible in the LMIC settings most burdened by enteric fever. Additionally, IgM and IgG ELISAs are intended for diagnosis based on paired acute and convalescent samples, which is impractical for immediate treatment decision-making [40]. If a single acute specimen is used for diagnosis, it can be difficult to rule-out a previous infection as being responsible for the observed titers. From single baseline specimens in this study, high IgM and/or IgG titers were found in 79.6% of confirmed enteric fever cases and 67.3% of probable cases. Given that serology was the only positive diagnostic test for probable cases, the high

titers observed could mislead clinicians toward an incorrect diagnosis if the titers resulted from a previous infection and not the current episode of illness.

Our study demonstrates the importance of evaluating available diagnostic tools for enteric fever during acute illness. The superior diagnostic accuracy of acute IgM and IgG ELISAs over TUBEX TF was particularly unexpected since sampling times for ELISA and TUBEX TF were similar. Optimization of testing approaches can improve diagnostic accuracy, disease management, and antibiotic stewardship. Alternative diagnostic methods with higher sensitivity and specificity are needed [42], such as detection of IgA responding to HlyE antigen [43], more sensitive molecular testing, improved antigen detection tests, and identification of specific biomarkers that can distinguish enteric fever from other infections. Blinding test operators to clinical data could also improve the objectivity of testing.

Our study had several limitations. First, this study was not designed to specifically evaluate enteric fever. Use of diagnostic assays varied, and comparison of assay results was not done using the exact same specimens. However, the time difference between samples taken for TUBEX TF and ELISAs was minimal, meaning that antibody levels should have been similar. Second, this study was not designed to evaluate TUBEX TF performance. The data was available because TUBEX TF is required by Indonesia's National Health Insurance system for enteric fever diagnosis, ensuring that every patient with clinical suspicion of enteric fever undergoes TUBEX TF testing. Our results may differ from those of studies designed *a priori* to study TUBEX TF in a controlled setting. Though generalizability of our findings may be limited, they are informative for health policy as they highlight the challenges of utilizing TUBEX TF for diagnosing enteric fever in the field. Third, antibiotic use data during hospitalization was not available from some sites because recording was not mandatory. Though clinicians were encouraged to obtain blood cultures prior to antibiotic administration, this was not a requirement, which could have resulted in negative blood cultures and PCR tests if appropriate antibiotic treatment was initiated quickly. Fourth, there is a possibility of recall bias on fever day. However, illness history was captured as early as possible after informed consent to reduce recall bias. Finally, samples positive for other pathogens did not always undergo testing for *Salmonella*. While the possibility of undetected co-infections cannot be ruled-out, this would be rare in an acute setting.

## Conclusion

Diagnosis of enteric fever remains challenging in Indonesia. The diagnosis is typically assumed for patients with acute febrile illness, though there is much overlap of signs and symptoms with other infections such as rickettsiosis and dengue fever. Accurate diagnostic tools are needed to properly diagnose and manage enteric fever. While TUBEX TF provides a convenient option for the assessment of infection, its performance characteristics are suboptimal in the Indonesian clinical setting and tend to over-diagnose enteric fever. We recommend strict adherence to TUBEX TF storage and processing instructions, as well as interpretation by technicians blinded to other clinical data, to improve the clinical utility of TUBEX TF. Additional evaluation of IgG from acute specimens for identification of enteric fever is merited, especially as the practicality of that approach improves. Alternative diagnostic methods for acute enteric fever should also be pursued.

## Supporting information

**S1 Table. Result of confirmatory tests on all enteric fever patients.**
(DOCX)

**S2 Table. Characteristics of correctly and incorrectly diagnosed enteric fever cases.**
(DOCX)

**S3 Table. Characteristics of cases with confirmed etiologies.**
(DOCX)

**S4 Table. Characteristics of patients with *Salmonella* Typhi and *Salmonella* Paratyphi A.**
(DOCX)

**S5 Table. Sensitivity and specificity of TUBEX TF, acute LPS IgG, and IgM by site.**
(DOCX)

**S6 Table. Sensitivity of TUBEX TF in confirmed and probable enteric fever cases.**
(DOCX)

**S7 Table. Sensitivity of TUBEX TF in *S.* Typhi and *S.* Paratyphi cases.**
(DOCX)

**S1 Fig. Days of tests performed for confirmed enteric fever cases.**
(TIF)

**S2 Fig. Percentage of positive tests performed for confirmed enteric fever cases.**
(TIF)

**S3 Fig. Kinetics of ELISA IgM and IgG in enteric fever and non-enteric fever cases.** Notes: The figure shows IgM and IgG results for every visit for all subjects. The heavy lines represent the median values of IgM and IgG for all *Salmonella* cases (black lines), pediatric *Salmonella* cases (red lines), adult *Salmonella* cases (blue lines), and all non-*Salmonella* cases (green lines). A loess smoother was applied in each instance.
(TIF)

**S1 Data. Data Enteric Fever for PLOS.**
(XLSX)

## Acknowledgments

We are grateful to the site staff at Hasan Sadikin Hospital (Bandung, Indonesia), Sanglah Hospital (Denpasar, Indonesia), Cipto Mangunkusumo Hospital (Jakarta, Indonesia), Sulianti Saroso Infectious Diseases Hospital (Jakarta, Indonesia), Wahidin Sudirohusodo Hospital (Makassar, Indonesia), Kariadi Hospital (Semarang, Indonesia), Soetomo Hospital (Surabaya, Indonesia), and Sardjito Hospital (Yogyakarta, Indonesia) for their contributions to this study. We also appreciate the technical support provided by the INA-RESPOND Secretariat. We are especially indebted to our study participants who made this research possible.

## Author Contributions

**Conceptualization:** Anggraini Alam, Tuti P. Merati, Pratiwi Sudarmono, Mansyur Arief, Usman Hadi, Abu Tholib Aman, Dewi Lokida, M. Hussein Gasem, Emiliana Tjitra, Aaron Neal, Herman Kosasih, Muhammad Karyana, Chuen-Yen Lau, Bachti Alisjahbana.

**Data curation:** C. Jason Liang.

**Formal analysis:** Syndi Nurmawati, C. Jason Liang.

**Funding acquisition:** Aaron Neal, Herman Kosasih, Muhammad Karyana.

**Investigation:** Syndi Nurmawati, Anggraini Alam, Hofiya Djauhari, Bachti Alisjahbana.

**Methodology:** Anggraini Alam, Tuti P. Merati, Pratiwi Sudarmono, Mansyur Arief, Usman Hadi, Abu Tholib Aman, Dewi Lokida, M. Hussein Gasem, Emiliana Tjitra, Aaron Neal, Herman Kosasih, Muhammad Karyana, Chuen-Yen Lau, Bachti Alisjahbana.

**Project administration:** Anggraini Alam, Tuti P. Merati, Pratiwi Sudarmono, Mansyur Arief, Usman Hadi, Abu Tholib Aman, Dewi Lokida, M. Hussein Gasem, Emiliana Tjitra, Aaron Neal, Herman Kosasih, Muhammad Karyana, Chuen-Yen Lau, Bachti Alisjahbana.

**Resources:** Syndi Nurmawati, Hofiya Djauhari.

**Software:** C. Jason Liang.

**Supervision:** Anggraini Alam, Tuti P. Merati, Pratiwi Sudarmono, Mansyur Arief, Usman Hadi, Abu Tholib Aman, Dewi Lokida, M. Hussein Gasem, Emiliana Tjitra, Aaron Neal, Herman Kosasih, Muhammad Karyana, Chuen-Yen Lau, Bachti Alisjahbana.

**Validation:** Anggraini Alam, Tuti P. Merati, Pratiwi Sudarmono, Mansyur Arief, Usman Hadi, Abu Tholib Aman, Dewi Lokida, M. Hussein Gasem, Emiliana Tjitra, Aaron Neal, Herman Kosasih, Muhammad Karyana, Chuen-Yen Lau, Bachti Alisjahbana.

**Visualization:** Syndi Nurmawati, Anggraini Alam, C. Jason Liang, Herman Kosasih, Bachti Alisjahbana.

**Writing – original draft:** Syndi Nurmawati, Bachti Alisjahbana.

**Writing – review & editing:** Syndi Nurmawati, Anggraini Alam, Hofiya Djauhari, Tuti P. Merati, Pratiwi Sudarmono, Vivi Setiawaty, Dona Arlinda, Retna Indah Sugiyono, Mansyur Arief, Usman Hadi, Abu Tholib Aman, Dewi Lokida, M. Hussein Gasem, Emiliana Tjitra, C. Jason Liang, Aaron Neal, Herman Kosasih, Muhammad Karyana, Chuen-Yen Lau, Bachti Alisjahbana.

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
