## [Decision Letter · Decision Letter 0]

23 Jan 2024

Dear Dr. Alisjahbana,

Thank you very much for submitting your manuscript "Clinical Characteristics of Typhoid Fever and Performance of TUBEX^®^TF IgM Test in Indonesian Hospitals" for consideration at PLOS Neglected Tropical Diseases. As with all papers reviewed by the journal, your manuscript was reviewed by members of the editorial board and by several independent reviewers. In light of the reviews (below this email), we would like to invite the resubmission of a significantly-revised version that takes into account the reviewers' comments. 

We cannot make any decision about publication until we have seen the revised manuscript and your response to the reviewers' comments. Your revised manuscript is also likely to be sent to reviewers for further evaluation.

Sincerely,

Kristen Aiemjoy

Guest Editor

Stuart Blacksell

Section Editor

Reviewer's Responses to Questions

**Key Review Criteria Required for Acceptance?**

**Methods**

-Are the objectives of the study clearly articulated with a clear testable hypothesis stated?

-Is the study design appropriate to address the stated objectives?

-Is the population clearly described and appropriate for the hypothesis being tested?

-Is the sample size sufficient to ensure adequate power to address the hypothesis being tested?

-Were correct statistical analysis used to support conclusions?

-Are there concerns about ethical or regulatory requirements being met?

Reviewer #1: -line 140, objective mentioned about the performance of Tubex TF only, where as results shows that performance of Tubex as well as Elisa. Thus objective should mentioned about the ELISA

- study design: appropriate for the study

- sample selection for validity of tubex T and LPS elisa not mentions is not clearly mentioned in the methods sections

- what is power of study with available sample size to measure the sensitivity and specificity?

Reviewer #2: Clear objectives

Appropriate study design

Population clearly described and appropriate

I think the sample size is sufficient

I think the correct statistical approach was used (I am not a statistician)

The appropriate ethical considerations have been documented

Specific comments

Line 172 In describing the reference laboratory testing, the methodology for the dengue, chikungunya, rickettsia and leptospirosis assays are not described. Would be valuable to include a short description or to reference the MH Gasem et al 2020 paper as the source of that information.

Line 176 It is not clear to me what antigen is used in the IgM and IgG ELISA assay. It would be helpful to know if it is the same as the O:9 lipopolysaccharide antigen used in the TUBEX TF assay.

Line 198 It would be helpful to include the definition of clinical typhoid used

Line 201 Bring the wording here into line with the wording used in Line 263 in results

Line 203 Suggest outlining with more clarity which participants had these extra tests

Reviewer #3: The objectives are clearly described although no hypothesis is stated for the performance of TUBEX. The study design is appropriate. There are no ethical concerns. 

Although the methods of the AFIRE study from which this study population were drawn have been published previously, some additional detail in the methods on the clinical standard of care for when TUBEX, Salmonella IgG and IgM ELISAs as well as the other diagnostic tests were performed would be helpful. Specifically, the relative timing of blood culture vs TUBEX vs the clinical diagnosis of enteric fever is not entirely clear. Do the clinicians use the blood culture and/or TUBEX results to make the clinical assessment? All the time? Some of the time?

Line 180: For the seroconversion—was this based only on the baseline (enrollment sample and the samples collected at 14-28ds (eg not including the 3m samples?) specify in the definition.

Was the differentiation of Typhi vs Paratyphi based on the blood culture/microbiology or was that only done by PCR? 

Line 185 says that PCR was done for non-Dengue patients with positive serology (is that positive in the acute sample or does that also include patients who seroconverted)? Later in the case definition (Line 200) & Fig 1 Notes, the Confirmed Salmonella cases are defined as positive blood culture AND/OR PCR. Were there any Confirmed cases that were blood culture negative and PCR positive? 

Since the TUBEX-TF test was developed specifically for S. Typhi, it may be worth at least conducting a sensitivity analysis where the ROC /AUC are calculated for Typhi and Paratyphi separately to determine whether there is any difference in test performance by subtype. At a minimum, the laboratory test results (eg similar to what's included in Fig 1) should be added to Table S3.

The Data analysis section should include some explanation for why the TUBEX cutoffs of >4 and >6 were chosen. 

Minor Comments:

Line 176: Specify that the antibodies detected by the ELISA kit are to S. Typhi LPS—if more detail is available from the manufacturer about the antigens please include.

Line 215-216: should read: …patient self-report which WAS categorized into sudden OR gradual…

**Results**

-Does the analysis presented match the analysis plan?

-Are the results clearly and completely presented?

-Are the figures (Tables, Images) of sufficient quality for clarity?

Reviewer #1: - what is dollar sign means in table 1- please mention in line 291-92

- There is discrepancy total salmonella mentioned in table 1 and S3?

- The table 1 is overcrowded, the author could remove the irrelevant information.

- The analysis plan mentioned about the sensitivity analysis however in result or discussion there is no mentioned about the finding of the sensitivity analysis

Reviewer #2: The analysis is appropriate and results presented clearly

Specific comments

Line 262-5 Does this need to be repeated as in the methods?

Difficult to read Figure 1 in the version received

Table S1 There were relatively few diagnoses on the basis of seroconversion (8+12) or twofold increase in titres (1+0) compared with elevated titres on baseline (43+37). Elevated baseline titres could be due to prior infection or cross reacting antibodies. This could be commented on in results or discussion.

SF1 In these figures showing the kinetics of the IgM and IgG serology over the three sample points, how were the solid summary lines derived?

Reviewer #3: The analysis generally matches the analytical plan. The results are presented accordingly, although because the different analyses include different patient sets, it is sometimes difficult to follow without repeatedly referring to the methods – particularly in the serologic testing section. There are also some discrepancies in the numbers between what is reported in the text and the various tables/figures that either need better explanation for the differences or need to be confirmed.

I am also confused by the choice to use the term ‘typhoid fever’ to refer to both S. Typhi and Paratyphi – it would be more accurate to refer to ‘enteric fever’. Similarly, the term ‘confirmed Salmonella’ is imprecise as there are many species of Salmonella; again ‘confirmed/probable enteric fever’ would serve.

The data analysis section of the methods section refers to a sensitivity analysis restricted to patients with both TUBEX and ELISA results – it is unclear where this is presented in the Results section.

Line 254: Provide some explanation for why the 22 subjects were not eligible.

Fig 1—why are the undiagnosed that are positive for Serology IgM or IgG not considered probable Salmonella?

Line 269-270: What was the clinician diagnosis for the patients subsequently determined to have Salmonella?

Line 278: specify – antibody indices at baseline.

Table 1 presents the clinical characteristics of physician diagnosed typhoid cases and Table S3 compares across pathogens regardless of physician diagnosis, but have the authors compared the clinical characteristics of the correctly diagnosed typhoid fever cases vs those that were incorrectly diagnosed? Did the cases that were incorrectly attributed have unusual clinical features? (the possibility also exists for co-infection, but that doesn't seem to have been assessed?) Although this is somewhat outside the scope of the authors' stated aims, it would be useful to understand factors that lead physicians to make the wrong diagnosis-- this could be accomplished by adding some columns to table S3 before the pooled 'All Salmonella' column

Line 305-306: This sentence is unclear about the findings re diarrhea. 

Line 317-318: The meaning of this is unclear -- is it that the time from symptom onset until testing was similar (as in when the blood was drawn that was tested? ) or the time delay between patient presentation and the availability of the laboratory results-- i think the former based on Table 2, but please clarify in the text.

I think it would help with the flow of the text if the last paragraph of the results -- eg the description of Fig S3 were moved to after table 2. Then all the remaining results will only be concerned with confirmed typhoid (eg not including the probable typhoid).

Line 314-315: the text says 91 participants tested in clinically non-typhoid fever group. But the denominators in Fig 1 in the TUBEX >=4 column in the bottom section add up to 109-- what is the explanation for this difference?

Similarly I am not sure who is included in the denominators in Table 3 for the TUBEX 134 confirmed non-Salmonella compared to the numbers presented in Fig 1. The control selection needs to be more clearly described (consider adding a section in the Methods after Case definitions).

Table 3: for the IgG it says 49 had >=1.1 but Table S1 suggests that only 44 were positive by IgG (30 with IgM and IgG > 1.1 + 11 with IgG > 1.1, IgM negative + 3 with IgM seroconversion, IgG high -- is that correct?) . The methods section and Table S1 use >1.1 rather than >= 1.1 as in Table 3. Is this a typo? If not, use a consistent threshold definition throughout the manuscript.

It would also be worth commenting somewhere in the text on whether there was any difference in the performance of the TUBEX for the probable Salmonella cases compared to the confirmed Salmonella-- eyeballing from Fig 1, it looks fairly similar. 

Minor comments

Line 273: Typhi should be capitalized

Table S3: Why is the total number of patients here 49 not 54?

Line 338-340-- this is in the table, but specify that this is based on the ELISA result only from the acute sample, not including seroconversions. 

Table 2: Consider shifting columns so that “Chikungunya” fits on one line; Probable Salmonella header is left aligned and the other headings are centered

Figure S3: Recommend splitting this into a 2 panel vertical figure rather than overlaying the lines and bars and having 2 y-axes -- it will be easier to interpret

**Conclusions**

-Are the conclusions supported by the data presented?

-Are the limitations of analysis clearly described?

-Do the authors discuss how these data can be helpful to advance our understanding of the topic under study?

-Is public health relevance addressed?

Reviewer #1: - In conclusion, author recommend about the strict protocol for storage and process as per leaflet, however author has not mentioned about the finding of problem of storge or process, neither author has discussed about this issue in detail. Thus this conclusion is not supported by the information presented.

Reviewer #2: I consider the conclusions supported by the data and the limitations have been described

The clinical and public health relevance of the results are addressed

Reviewer #3: The conclusions are generally supported by the data presented and the public health relevance is addressed. 

Lines 388-390: It is curious that the authors raise this limitation when it seems like they have access to the data to perform the analysis of/comparison to patients not diagnosed with typhoid fever.

Line 432-433: I'm not sure if the ELISA kit used in this study is the same LPS, but Aiemjoy et al report on longitudinal kinetics of anti-S. Typhi LPS IgG. doi: 10.1016/S2666-5247(22)00114-8

Line 451-452: given this statement, it would be useful to provide (in the methods) details on who performed the TUBEX tests (hospital laboratory staff?) and whether they knew the provisional diagnosis of the patient.

Lines 459-461: Recommend moving this information about the National requirement to test with TUBEX to the end of the Introduction because it helps provide context for the study aims. 

Minor comments

Line 438: should read ...and the need to DO batch testing...

Line 469: Italicize Salmonella

**Editorial and Data Presentation Modifications?**

Reviewer #2: Outlined in methods and results

Reviewer #3: Figure S2 should be included in the main body of the paper.

Consider using term “low and middle income countries” or “typhoid affected countries” rather than the outdated term “developing countries.”

Abstract methodology section mentions AFIRE study at eight “local hospitals.” Could a better description be used like “eight Indonesian hospitals” or “eight hospitals in [location]”?

Introduction Lines 121 - 124 may want to also cite the more recent evaluation by Sapkota et al doi: 10.1128/jcm.01000-22

Introduction: Line 130: Consider adding sentence describing mechanism of TUBEX, potentially moving the descriptive sentence from Lines 166-168.

**Summary and General Comments**

Reviewer #1: The study present the performance of TUBEX TF in detecting the Typhoidal Salmonella infection using the culture or PCR positive typhoid infection or other alternate etiology in existing health care delivery system and showed the difference in the real world performance in comparison to controlled study envi

---

## [Decision Letter · Decision Letter 1]

8 May 2024

Dear Dr. Alisjahbana,

Thank you very much for submitting your manuscript "Clinical Characteristics of Enteric Fever and Performance of TUBEX TF IgM Test in Indonesian Hospitals" for consideration at PLOS Neglected Tropical Diseases. As with all papers reviewed by the journal, your manuscript was reviewed by members of the editorial board and by several independent reviewers. The reviewers appreciated the attention to an important topic. Based on the reviews, we are likely to accept this manuscript for publication, providing that you modify the manuscript according to the review recommendations. 

Two additional comments noted by the editor: 

1. The order of the specificities in the abstract (line 70) is reversed

2. The subtext of Table 3 should be corrected to something like test positive/confirmed positive for sensitivity and test negative/confirmed negative for specificity

Sincerely,

Kristen Aiemjoy

Guest Editor

Stuart Blacksell

Section Editor

Two additional comments noted by the editor: 

1. The order of the specificities in the abstract (line 70) is reversed

2. The subtext of Table 3 should be corrected to something like test positive/confirmed positive for sensitivity and test negative/confirmed negative for specificity

Reviewer's Responses to Questions

**Key Review Criteria Required for Acceptance?**

**Methods**

-Are the objectives of the study clearly articulated with a clear testable hypothesis stated?

-Is the study design appropriate to address the stated objectives?

-Is the population clearly described and appropriate for the hypothesis being tested?

-Is the sample size sufficient to ensure adequate power to address the hypothesis being tested?

-Were correct statistical analysis used to support conclusions?

-Are there concerns about ethical or regulatory requirements being met?

Reviewer #1: Objectives are clearly articulated but second objectives " evaluate performance of TUBEX TF" should be evaluate perfornace of TUBEX TF and Ig M and IgG ELISA as result in table 3 shows sensitivity and specificity of both against blood culture or pcr.

-study design is appropriate to address the stated objectives

- Population is clearly described 

- all ethical and regulatory requirements are met

- sensitivity analysis done for independent variable should be separated from sensitivity analysis done for validation of the test

Reviewer #2: The study design and methods are clearly described. The study includes an appropriate population. THe analysis was appropriate

Reviewer #3: The methods now clearly describe the study and analyses presented in the manuscript.

**Results**

-Does the analysis presented match the analysis plan?

-Are the results clearly and completely presented?

-Are the figures (Tables, Images) of sufficient quality for clarity?

Reviewer #1: - Results are clearly presented and tables and figures are clear

Reviewer #2: The results, tables and figures are clearly presented

Reviewer #3: The results are now connected to the analysis plan and clearly described. The tables and figures are of sufficient quality. 

Minor comments:

- Figure S3 -- it would be good to include a legend that explains that the heavy solid lines are means (was a loess smoother used?)

**Conclusions**

-Are the conclusions supported by the data presented?

-Are the limitations of analysis clearly described?

-Do the authors discuss how these data can be helpful to advance our understanding of the topic under study?

-Is public health relevance addressed?

Reviewer #1: - The conclusion is appropriate

- the author has nicely discussed about the study finding as well as described the limitation of the study

Reviewer #2: The conclusions are approopriate and placed in a ublic helath context

Reviewer #3: The conclusions are supported by the data presented and the limitations are described. 

Minor comments:

-Line 490 -- I believe there is an extraneous period before the references (34,35) 

-Line 510-512 -- I suggest moving the section on sensitivity before the paragraph on specificity (or integrating it). However, I don't think you are advocating for using a >=4 cutoff given the poor specificity, so you may want to consider this language a bit more.

-- Line 559 -- should read ...to study TUBEX TF in A controlled setting.

**Editorial and Data Presentation Modifications?**

Reviewer #1: Accept with minor revision

Reviewer #2: None suggested

Reviewer #3: (No Response)

**Summary and General Comments**

Reviewer #1: the paper has nicely articulated now and discuss on the important topic of real world utility of alternative diagnostic for enteric fever TUBEX TF and ELISA in addition to current gold standard, blood culture.

Reviewer #2: The authors have kindly addressed previous review comments. I have no further suggestions for this revised version

Reviewer #3: The clarity of the manuscript is much improved and the additional clinical detail is beneficial. With a few additional mostly copy edits, I believe the paper is ready for publication.

Minor comments 

In a few places the replacement of typhoid with enteric fever is incomplete (line 58, 119)

-Line 92: should read ...enteric fever cases HAS been.. (the use of clinical data...has been considered.)

-Line 136-138 I think this sentence should precede the sentence about the Sapkota manuscript because I think it is describing the studies included in the meta analysis referenced in line 133?

PLOS authors have the option to publish the peer review history of their article (what does this mean?). If published, this will include your full peer review and any attached files.

Reviewer #1: No

Reviewer #2: No

Reviewer #3: No

Figure Files:

Data Requirements:

Reproducibility:

References

---

## [Editor Report · Decision Letter 2]

26 Jun 2024

Dear Dr. Alisjahbana,

We are pleased to inform you that your manuscript 'Clinical Characteristics of Enteric Fever and Performance of TUBEX TF IgM Test in Indonesian Hospitals' has been provisionally accepted for publication in PLOS Neglected Tropical Diseases.

Best regards,

Kristen Aiemjoy

Guest Editor

Stuart Blacksell

Section Editor

---

## [Editor Report · Acceptance letter]

19 Jul 2024

Dear Dr. Alisjahbana,

We are delighted to inform you that your manuscript, "Clinical Characteristics of Enteric Fever and Performance of TUBEX TF IgM Test in Indonesian Hospitals," has been formally accepted for publication in PLOS Neglected Tropical Diseases.

Best regards,

Shaden Kamhawi

co-Editor-in-Chief

Paul Brindley

co-Editor-in-Chief
